# Meticulous and Early Understanding of Congenital Cranial Defects Can Save Lives

**DOI:** 10.3390/children10071240

**Published:** 2023-07-18

**Authors:** Ali Al Kaissi, Sergey Ryabykh, Farid Ben Chehida, Hamza Al Kaissi, Susanne Gerit Kircher, Franz Grill, Alexander Guben

**Affiliations:** 1National Medical Research Center for Traumatology and Orthopedics n.a. G.A. Ilizarov, 640014 Kurgan, Russia; rso_@mail.ru; 2Ibn Zohr Institute of Diagnostic Radiology, Cite Al Khadra, Tunis 1003, Tunisia; if.chehida@gnet.tn; 3Clinic for Dermatology and Allergology, Luisen Hospital, 52064 Aachen, Germany; alkaissihamza@gmail.com; 4Center of Medical Patho-Biochemistry and Genetics, Medical University of Vienna, 1090 Vienna, Austria; 5Pediatric Department, Orthopedic Hospital of Speising, 1130 Vienna, Austria; 6Department of Orthopedic Department, Saint-Petersburg State University Hospital, 199034 St. Petersburg, Russia

**Keywords:** marfanoid habitus, craniosynostosis, Shprintzen–Goldberg syndrome, increased intracranial pressure, MRI, CT scan

## Abstract

Background: Omitting the early closure of the cranial sutures in newly born children is not an uncommon practice. We describe the natural history of several unrelated children and adults from two unrelated families. These children were born with variable clinical manifestations: craniofacial asymmetry, ocular proptosis, floppiness, and progressive deceleration in cognitive development. None of these children underwent a cranial sutures assessment. False diagnoses of positional plagiocephaly, neonatal thyrotoxicosis, congenital muscular atrophy, and hydrocephalus were given to the parents. This sort of malpractice was the reason behind a sequence of devastating pathological events that occurred in the lifetime of these children and adults. Material and Methods: This was a multigenerational study of two unrelated families. In total, we studied three children (aged 7–19 years) and three adults (aged 40–52 years) from two families. The children from the first family were referred to our departments because of pre-pubertal scoliosis, kyphoscoliosis, and early-onset osteoarthritis. Reading the clinical histories of these children signified apparent clinical misconceptions. For instance, craniofacial asymmetry was misinterpreted as positional plagiocephaly and treated by means of helmet molding therapy. Ocular proptosis was given the false diagnosis of neonatal thyrotoxicosis. Floppiness (hypotonia) was misdiagnosed as congenital muscular dystrophy. The index case from the second family showed progressive deceleration in his cognitive development, associated with signs of increased intracranial pressure. The only diagnosis was Dandy–Walker malformation. We documented every patient in accordance with the clinical and radiological phenotypic characterizations. The genotype characterization followed accordingly. Results: All patients in family (I) manifested a phenotype consistent to a certain extent with the clinical phenotype of Shprintzen–Goldberg syndrome (SGS), though the intensity of spine deformities was greater than has been described in the literature. The second family showed a constellation of Marfanoid habitus, craniosynostosis, increased intracranial pressure, hydrocephalus, Dandy–Walker malformation, seizures, and intellectual disability. The overall clinical phenotype was consistent but not fully diagnostic of craniosynostosis–Dandy–Walker-malformation hydrocephalus syndrome. The early closure of the sutures was totally different from one patient to another, including the premature closure of the metopic, coronal, squamosal, and sagittal sutures. One patient from family (II) underwent the implementation of a shunt system at the age of 3 years, unfortunately passing over the pre-existing craniosynostosis. In addition to skeletal deformities, a history of seizures and severe intellectual disability was recorded. The proband underwent chromosomal karyotyping, the FISH test, and whole-exome sequencing. Conclusion: The purpose of this study was fivefold. Firstly, to gain a meticulous understanding in order to differentiate between positional plagiocephaly, hypotonia, and congenital exophthalmos and their connections to abnormal craniofacial contours was and still is our first and foremost concern. Secondly, we aimed to characterize craniosynostosis, seizures, intellectual disabilities, and hydrocephalus associated with Marfanoid habitus, which were clearly demonstrated in our patients. Thirdly, we aimed to address the imperative for interpretations of clinical and radiological phenotypes and relate these tools to etiological understanding, which is an essential basis for diagnosis in the majority of long-term pediatric admissions. Fourthly, we aimed to assess the impacts of the missed early closure by the pediatricians and pediatric neurologists, which added a heavy pathological burden on these patients and their families. Fifthly, we aimed to identify whether early and diligent recognition can assist in cranial vault remodeling via surgical intervention to halt premature cranial suture fusions and can possibly alter the devastating course and the complications of the synostosed sutures.

## 1. Introduction

Craniosynostosis is the early closure of cranial sutures, which eventually leads to the maldevelopment of the craniofacial contour. Omitting early diagnosis results in a series of devastating complications. The first and the foremost adverse outcome is irreversible impairment of the normal growth and development of the brain, as well as disruption to the craniofacial anatomy. In a newborn infant, the membranous bones of the skull vault are separated by articulated anatomical structures called the sutures. These flexible articulations facilitate the passage of the baby through the birth canal. Interestingly, the sutures give the growing and developing brain the opportunity to expand in a normal, directive genetic strategy. Unfortunately, when one or more of these sutural articulations is closed prematurely, the outcome for the growing brain is drastic. The growing brain, when facing resistance from the synostosed suture, will search for another permitting flexible/open sutural articulation, resulting in the development of a pathological mechanism that leads to variable degrees of irreversible brain damage correlated with the number of early closed sutures, and ending with substantial distortion of the craniofacial features [1,2,3,4]. 

Shprintzen–Goldberg syndrome (SGS) is another form of craniosynostosis. SGS is characterized by craniosynostosis, ocular proptosis, maxillary and mandibular hypoplasia, prominent lateral palatine ridges, low-set soft ears, abdominal hernias, and arachnodactyly. Shprintzen–Goldberg syndrome (OMIM #182212) is an autosomal dominant disorder, characterized by a constellation of clinical features such as a Marfan-like habitus, premature closure of the cranial sutures, craniofacial asymmetry, skeletal deformities, seizures, and intellectual disability [5,6,7,8]. Mutations in the transforming growth factor beta (*TGF-β*) signaling pathway have been shown to play a major role in the etiology in a remarkable number of patients with variable syndromic entities, such as Loeys–Dietz syndrome (LDS) and Shprintzen–Goldberg syndrome (SGS). LDS (types 1–5, OMIM# 609192, 610168, 613795, 614816, 615582) is characterized by an autosomal dominant pattern of inheritance with the involvement of many systems. LDS results from heterozygous mutations in the *TGF-β* signaling pathway [9]. Genetic mutations in exon 1 of *SKI* have recently been described as being connected to approximately 90% of reported individuals clinically diagnosed with Shprintzen–Goldberg syndrome (SGS) [10,11]. The genetic etiology of hydrocephalus consists of a long list of syndromic associations. In the vast majority of cases, hydrocephalus is just a symptom complex rather than a definite diagnostic entity [12]. Hydrocephalus is mostly diagnosed in patients with syndromic entities, with lesser incidence in patients with non-syndromic types of craniosynostosis. The main pathological process in syndromic type is almost correlated with primary cerebral ill-development, which eventually leads to structural hurdles for the hydrodynamics of the cerebrospinal fluid (CSF) [13,14]. Bradock et al. described the association of the malformation complex of Dandy–Walker and sagittal craniosynostosis [15]. 

## 2. Material and Methods 

The study protocol was approved by the Ethics Committee of the Ilizarov Scientific Research Institute, No. 4(50)/13.12.2016, Kurgan, Russia. Informed consent was obtained from the patients’ guardians. We fully documented children and adults through detailed clinical and radiological phenotypic characterizations at the Osteogenetic Outpatient Clinic at the Orthopedic Hospital of Speising (pediatric department), Vienna, and through the scientific collaboration of the first author with the Ilizarov Center, Kurgan, Russia, and with the Ibn Zohr Institute of Diagnostic Radiology, Tunis. Clinical documentation and the detailed study of several family subjects from two unrelated families were carried out to understand, anticipate, and interpret the clinical complexity among variable family subjects. We emphasized maternal clinical history through analyzing the circumstantial unusual events that occurred through the antenatal, prenatal, and postnatal courses of gestation. Maternal gestational history encompasses the history of multiple/spontaneous miscarriages, especially in the first trimester, the nature of the fetal movement in utero (hypoactive/hyperactive or hypertonic), any history of stillbirths, premature delivery, breech, or any other unusual presentation, and the revision of ultrasound images and, if possible and available, the ultrasound CD.

Neonatal history includes the apgar score and notes records of weight, length and head circumference. History of cyanosis, respiratory distress syndrome, fetal resuscitation, convulsions, and the time spent in the intensive care baby unit are also included. Developmental history is a baseline tool required in building up the diagnostic process. A family study includes the clinical phenotype of the parents, siblings, and relatives, as much as possible. Family photo albums are an essential part of the diagnostic process. The structured questionnaire is based on detecting the history of apnea, migraine, stroke, aneurysm, and sudden/unexplained death, any history of learning disabilities, seizures, brain tumors, and so forth. For the pediatric group, the assessment of pre- and postnatal growth curves (growth acceleration or deficiency) is of high concern. Biomechanical examination includes the craniocervical, axial, and appendicular systems, accordingly. It is imperative to examine the skin in patients with Marfan-like phenotype, especially in patients with other forms of connective tissue disorders. Patients with skin stigmatas as seen in hamartoneoplastic disorders can also manifest Marfan-like disorder. In this study, we explored the clinical and radiological phenotypic characterizations in three children (aged 7–19 years) and three adults from two unrelated families (aged 40–52 years). The children were referred to our departments because of pre-pubertal scoliosis, kyphoscoliosis, antalgic gait, and early osteoarthritis. Reading the clinical histories of these children with care showed that immediately after birth, different misdiagnoses were given. Certain clinical features were misinterpreted by pediatricians, which resulted in a cluster of dreadful complications. 

The family study of each child showed variable forms of sutural synostosis from one family subject to another. Interestingly, several family subjects did not show classical Marfan-like features, as manifested in the probands, but exhibited diverse forms of skeletal and visceral abnormalities. 

We described our patients in accordance with the clinical presentation and the definite diagnosis. The family genealogy search included the comprehensive clinical examination of the siblings, parents, and close relatives. There were two families, in which marfanoid habitus associated with craniosynostosis were the major clinical findings. The clinical and radiological phenotypic characterizations were our prime line tools, followed by genetic confirmation. 

## 3. Results 

### 3.1. Family I: Shprintzen-Goldberg Syndrome-like Results with Severe Spine Tilting 

The first subject was an 11-year-old boy. He was the product of a full-term pregnancy, and was born to a 26-year-old woman, and an unrelated 31-year-old man. The mother, prior to giving birth to her first child, had experienced multiple spontaneous abortions in her first trimester (more than four times). The child was born with floppiness (hypotonia). The pediatric neurologist gave a diagnosis of congenital muscular atrophy. From his neonatal period and through the first seven months, he underwent a series of neuromuscular investigations. His severe ligamentous hyperlaxity and poor muscle mass were confusing clinical parameters for the pediatric neurologist to establish the misdiagnosis of congenital muscular dystrophy. Vigorous investigations were implemented. His EMG (electromyography tests) showed minor myopathy changes. Similarly, his past muscle MRI showed some nonspecific, and non-diagnostic changes. The pediatric neurologist suggested genetic screening for gene-related myopathy, which finally proved negative, and the muscle biopsy yielded normal results. His subsequent course of development was one of marked developmental delay, especially in his gross motor, fine motor, and speech skills. His first visit to the department of rheumatology was at the age of four years, when his parents observed flat feet and bouts of pain in his ankle and knee joints (early osteoarthritis of the weight bearing joints). These were treated by the rheumatologist by prescribing certain types of ointments. At the age of 10 years, spinal malalignment was recognized. At that time, kyphoscoliosis was recognized and the child was treated through the application of a Milwaukee brace. Then, the child was referred to the first author for the further clarification of his clinical status. The child showed an abnormal craniofacial contour with a clear-cut clinical picture of Marfanoid features. He had facial asymmetry, with a large and prominent frontal area. Despite his marfanoid habitus of long thin limbs and arachnodactyly, his height was around the 50th percentile. A musculo-skeletal examination showed generalized ligamentous hyperlaxity associated with joint hypermobility. Through the application of the Beighton scoring system, he manifested bilateral full and passive dorsiflexion of the little finger, beyond 90°. Thumbs showed bilateral full passive dorsiflexion to the flexor aspect of the forearm. He showed bilateral hyper-extension of the elbows of 15°. The forward flexion of the trunk with knees fully extended showed that the palms and hands could rest flat on the floor. Hyper-extension of the knees beyond 10° was confirmed, as well. The examination of the skin showed minor signs of hyper-elasticity. All in all, his clinical status was compatible with nine out of nine points in accordance with the Beighton scoring system (this confirms the generalized ligamentous hyperlaxity and joint hypermobility), which also explained the reason for his flat feet and the early onset osteoarthritis.

Clinical documentation and radiological documentation are shown in Figure 1a,b. The clinical photo showed craniofacial dysmorphic features, marfanoid habitus, long and thin limbs, arachnodactyly, intellectual disability, and scoliosis of 65° of Cobbs angle, and a Milwaukee brace was prescribed to prevent further spine tilting (Figure 1a). The lateral skull radiograph showed scaphocephaly (the skull is of abnormal contour, showing a bilateral narrow head because of premature closure of the sagittal suture). The synostosed sagittal suture formed a well-palpated osseous ridge (Figure 1b). 

Molecular genetics: Chromosomal karyotyping and the FISH test was applied and structural chromosomal aberrations were excluded with 20 CAG-banded mitoses. There were no microdeletions or micro duplication after performing array-CGH analysis. No mutations in the FBN1, TGFBR1, and TGFBR2 genes were encountered.

The family history revealed that a younger male sibling was born full-term and was 7 years old at the time of study. In his neonatal period, his pediatrician observed ocular proptosis associated with hypotonia. Ocular proptosis has been confused with exophthalmos. A diagnosis of neonatal thyrotoxicosis was given to the family. The pediatrician based his diagnosis on the maternal history of spontaneous abortions prior to the gestation of the two surviving children. However, neither the mother nor this child showed any clinical or laboratory features suggestive of thyrotoxicosis. The child was extensively investigated to rule out the diagnosis of neonatal thyrotoxicosis. Despite the prenatal investigations during gestation, none of the signs of neonatal thyrotoxicosis were elicited. No fetal or neonatal tachycardia was recorded. His resting fetal heart rate was normal, and no fetal neonatal goiters were recorded. Afterwards, the diagnosis of positional plagiocephaly was issued and the pediatricians suggested the application of “Helmet molding therapy” because of the skull asymmetry. Helmet molding therapy was applied for two years. The incentive for examining this child was as part of our clinical strategy for examining all the family subjects. This child was seen for the first time by the first author at the age of 7 years. He manifested typical marfanoid habitus akin to his older sibling. This can be seen through the clinical documentation, as summarized via his images. Magnetic resonance imaging (MRI) showed the ocular proptosis (arrows) in his neonatal period, which was misdiagnosed as neonatal thyrotoxicosis (Figure 2a). At the age of 7 years, he was referred to our department to check his juvenile scoliosis. He manifested torticollis and cervico-thoracic kyphoscoliosis of 50° kyphoscoliosis. He showed a similar clinical phenotype of marfanoid habitus, craniofacial asymmetry, and dysmorphic craniofacial features, suggestive of Shprintzen Goldberg syndrome (SGS) (Figure 2b). A 3D reconstruction computerized tomography (CT) scan showed craniofacial asymmetry and scaphocephly in connection with the total fusion of the sagittal suture (arrow) (Figure 2c). 

The clinical examination of the parents revealed a very interesting outcome. The mother showed a typical clinical and radiological phenotype, as seen in her two offspring. She manifested marfanoid habitus associated with craniosynostosis. The mother was 40 years old and showed a similar typical clinical phenotype as her two siblings. A 3D reformatted skull CT scan showed totally fused metopic, coronal, and sagittal sutures, which resulted in very easily palpable ridges over the synostosed sutures. There was a noticeable disproportion between the cranium and the facial bones. In other words, this disproportion between the cranial and facial bones’ growth resulted in a huge difference because of the overgrowing of the facial bones on account of the cranial bones (Figure 3a). The reformatted skeletal phenotype CT scan of the mother showed a typical marfanoid habitus, long and thin limbs, long trunk and large cranium, macrognathia, and cervico-thoracic scoliosis of 70° Cobb’s angle. In her adolescence, her scoliosis did not respond to corset treatment and she refused surgical intervention to re-align her spine (Figure 3b) (please refer to Table 1).

### 3.2. Family II: Marfanoid Habitus, Craniosynostosis–Dandy–Walker Malformation-Hydrocephalus Syndrome

The index case is an 18-year-old Austrian boy who presented with a tall stature (97th percentile), long arms and legs and relative arachnodactyly. An abnormal craniofacial contour resulted in a long, narrow head (prominent forehead and narrow temples), mid-facial hypoplasia and thick, dense eyebrows. A clinical examination of the skull showed apparent and elevated bony ridges, more obvious bilaterally along the squamosal sutures, as well as along the sagittal suture. A family history showed that the 52-year-old father manifested marfanoid habitus (97th percentile). The parents revealed that the child´s subsequent developmental history was near-normal, especially in his first two years. After the age of two years he started to show an awkward gait, and poor balance associated with psychosomatic impairment and signs of increased intracranial pressure. At that age he underwent a series of profound investigations. Serum *Chromogranin* A (a serum marker of neuroendocrine neoplasia), ammonia, and uric acid, were all within normal values. There were no signs of organic acidopathy, aminoacid excretion pattern, and no increase in MPSs excretion shown through screening for mucopolysacharidosis in urine. Whole exome sequencing did not reveal either Fragile X syndrome nor Dandy–Walker as a syndromic entity. An MRI showed enlarged ventricles caused by a Dandy–Walker-like malformation. The neurosurgeon suggested performing a shunt operation via a surgical perforation of the Blake’s pouch cyst. Unfortunately, the shunt operation altered nothing and the progressive intellectual deterioration became worse. We examined the detailed analysis of the three-dimensional CT scan for both the ecto- and endocranial surfaces of the skull. The scout CT scan of the skull showed a clear scaphocephaly, which signifies the early fusion of the sagittal suture (Figure 4a). The 3D reconstruction CT scan of the skull showed quite clear apparent premature neurocranial suture closure, observed along the squamosal sutures (arrows), and associated with the partial closure of the coronal sutures (there were apparent bony ridges of bilateral squamosal sutures consistent with premature craniosynostosis) (Figure 4b). Coronal computed tomography images of the brain demonstrated a local thumb printing/beaten copper appearance (arrows) of the inner cortex in the region signifying bilateral squamosal fusion (white arrows) (Figure 5a). The early sutural fusion extended posteriorly from the pterion and connected the temporal squama with the inferior border of the parietal bone (bulge-like in the lateral view) (arrow heads). Note the early fusion of the mendosal suture (arrow) (known as the accessory occipital suture) and the implanted shunt (Figure 5b). 

## 4. Discussion

Skull malformation complexes are not uncommon clinical presentations in pediatric departments. Pediatricians bear full responsibility for recognizing and identifying abnormal craniofacial changes [16,17]. Positional plagiocephaly has been overly misused in most pediatric disciplines, thereby giving access to the mismanagement of the vast majority of infants and children with craniosynostosis [18,19]. Ventricular dilatation in the presence of early cranial sutural closure is a unique condition in correlation with the pathologic process and the final clinical phenotypic characteristics. It is not common in non-syndromic craniosynostosis, and in such conditions it is usually traceable to coexisting hidden disorders. Strikingly, ventricular dilatation is not an uncommon complication seen in several craniosynostosis disorders. It can be encountered in patients with Crouzon’s, Pfeiffer’s, or Apert syndrome. Furthermore, shunt-dependent hydrocephalus is a well-known procedure applied for Crouzon’s and in children with Pfeiffer syndrome [20,21,22]. The clinical phenotype of Shprintzen–Goldberg syndrome (SGS) somewhat mimicks Marfan syndrome. SGS leads to the development of a diversity of skeletal and visceral dreadful abnormalities due to the generalized involvement of the connective tissues. The premature closure of the cranial sutures is the reason for the development of ocular proptosis and the distortion of the normal anatomy of the facial bones. Dysmorphic facial features are variable and range between prominent and wide frontal areas to low-set ears and a small mandible. Generalized ligamentous hyperlaxity is the cause of antalgic gait and early osteoarthritis. Milestones may be delayed and hypotonia with mental retardation has been encountered. Patients become increasingly dysmorphic with age, with marked hypertelorism, shallow orbital ridges, and downslanting palpebral fissures, and marked micrognathia in some. The complications are vast, such as intellectual disability, seizures, scoliosis, kyphosis, and Scheuermann’s disease [5,23,24]. 

Dandy–Walker malformation (DWM) is a well-known human cerebellar malformation, characterized mainly by defective development of the cerebellar vermis, cystic lesion of the fourth ventricle, and an enlarged posterior fossa with upward displacement of the lateral sinuses, and tentorium. There are several genetic loci described in the literature related to DWM, as well as syndromic associations [25,26]. Hydrocephalus is strongly correlated with the development of clusters of symptoms of increased intracranial pressure, as well as neurological deficits in patients with DWM. A proper embryological understanding of the hindbrain is the cornerstone for understanding cerebellar malformations, including DWM and other related entities. Surgical interventions for children with premature closure of the sutures have to be performed in the first few months of life. The management of abnormal craniofacial contours is basically directed to reshape the defective cranial bones, aiming to give ample opportunity for normal brain growth. Sadly, omitting the first early alarming signs of premature closure of the sutures leads to progressive skull deformation and the persistence of the complications. In other words, it is the worst-case scenario, and the presumed optimistic outcome of the operations can barely be considered. It becomes more difficult to carry out cranial corrections in children as they get older, which is associated with severe, difficult-to-control complications [27]. Abnormal craniofacial contours stemming from the premature closure of the cranial sutures leads to the suppression of brain growth, and defective dynamics of cerebrospinal fluid (CSF) result in increased intracranial pressure, hydrocephalus, and a constellation of neurological deficits. Therefore, early diagnosis and craniectomy is preferably performed in children who manifest a premature closure of several sutures. Craniectomy aims to give the brain the best chance for normal expansion, and this is performed to ameliorate the cranial shape and to minimize the possibility of re-closure of the sagittal suture [28,29]. 

## 5. Conclusions 

Omitting the early closure of the cranial sutures in our current patients was the reason for the development of a long list of complications. Some experts erroneously might think that if we cannot cure genetic diseases, we can only detect them by using sophisticated and expensive laboratory technology. Others claim that the diagnosis of disability can only be accomplished through intensive biological testing. In fact, we were able to diagnose hundreds of children and adults via meticulous and comprehensive clinical and radiological phenotypic characterizations. We believe that treating disabilities mainly involves our sense of humanity and our professional honesty. For most of us, these diseases do not evoke anything, and for good reason they are all part of what has been called and falsely quoted as orphan diseases or rare diseases. It is true that the severe types of syndromes are rare and manifest themselves soon after birth, but the less severe forms mostly become common and exhibit various skeletal and extra-skeletal deformities in adolescence or later in life.

We can confirm that the term rare is used when there is a lack of knowledge and experience among the medical staff, and reflects the immense failure of the clinicians in connecting the unusual, mild, or moderate abnormalities in different family subjects which are relevant to the syndrome. Many of these disorders are so puzzling and disconcerting that they intimidate many pediatricians. Therefore, a tangible understanding of the complexity of these diseases on the part of our colleagues is a perquisite.

## Figures and Tables

**Figure 1 children-10-01240-f001:**
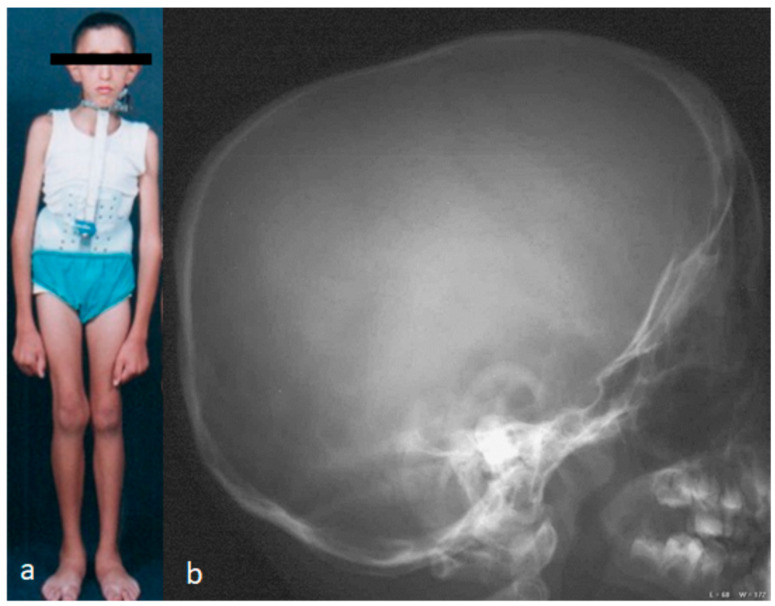
(**a**,**b**). The clinical phenotype of an 11-year-old child referred to the senior author with a diagnosis of pre-pubertal idiopathic scoliosis. Clinically, he manifested craniofacial dysmorphic features, marfanoid habitus, long and thin limbs, arachnodactyly, intellectual disability, and scoliosis of 65° of Cobbs angle (**a**). Lateral skull radiograph showed scaphocephaly (the skull is of abnormal contour and appeared narrow because of premature closure of the sagittal suture). The sagittal suture formed a well-palpated osseous ridge (**b**).

**Figure 2 children-10-01240-f002:**
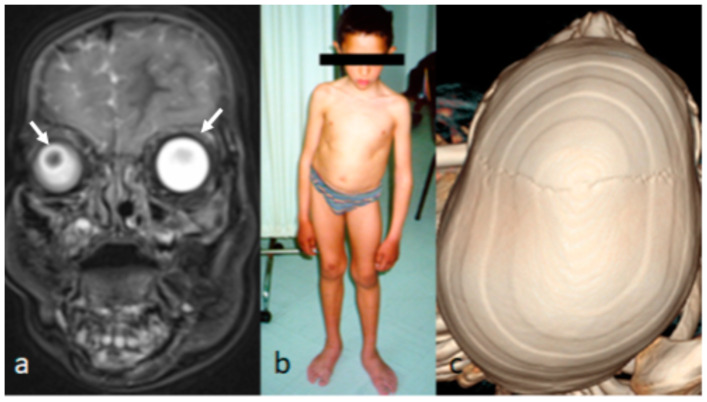
(**a**–**c**). MRI showed the ocular proptosis (arrows) of a newborn which was misdiagnosed as neonatal thyrotoxicosis. The craniofacial asymmetry was totally ignored by the radiologist and the pediatrician (**a**). The clinical phenotype of the younger sibling at the age of 7 years. He was referred to our department to check his juvenile scoliosis. He manifested torticollis and cervico-thoracic kyphoscoliosis of 50° kyphoscoliosis. He showed a similar clinical phenotype to his older sibling (craniofacial asymmetry and dysmorphic craniofacial features suggestive of Shprintzen–Goldberg syndrome (SGS) (**b**). A 3D reconstruction CT scan showed craniofacial asymmetry and scaphocephly in connection with the premature fusion of the sagittal suture, and the application of helmet therapy enhanced the cranial asymmetry (arrow) (**c**).

**Figure 3 children-10-01240-f003:**
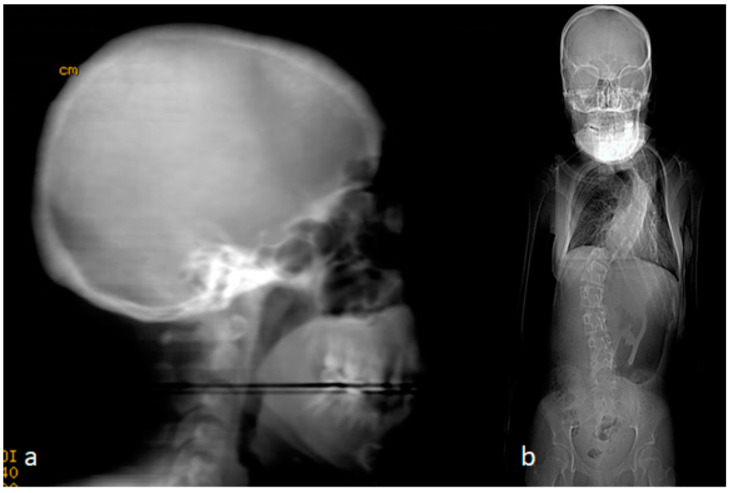
(**a**,**b**). A 40-year-old mother with the typical clinical phenotype of her two siblings. A 3D reformatted skull CT scan showed totally fused metopic, coronal, and sagittal sutures which resulted in very easily palpable ridges over the synostosed sutures. There was a noticeable disproportion between the cranium and the facial bones. In other words, the stoppage of the cranial bone growth resulted in a huge difference from the overgrowing of the facial bones (**a**). The reformatted skeletal phenotype CT scan of the mother showed typical marfanoid habitus, long limbs, long trunk, small cranium, large facial bones, macrognathia, and cervico-thoracic scoliosis of 70° Cobbs angle. In her adolescence, her scoliosis did not respond to corset treatment and she refused surgical intervention to re-align her spine (**b**).

**Figure 4 children-10-01240-f004:**
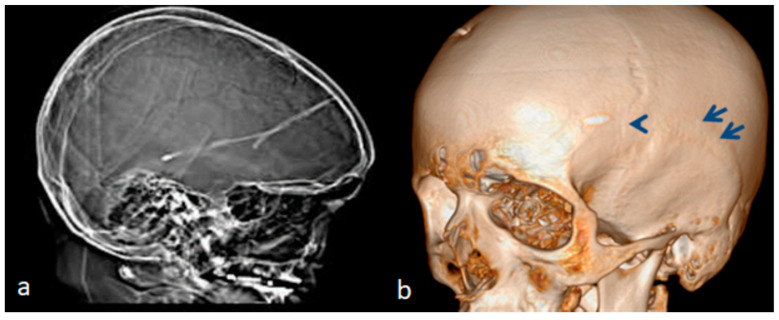
(**a**,**b**). Scout CT scan of the skull showed a clear scaphocephaly which signifies early fusion of the sagittal suture (Figure 4a). The 3D reconstruction CT scan of the skull shows quite clear apparent premature neurocranial suture closure easily observed along the squamosal sutures (arrows) associated with partial closure of the coronal sutures (there are apparent bony ridges of bilateral squamosal sutures consistent with premature craniosynostosis). The bony ridges along the squamosal suture arching posteriorly from the pterion and connecting the temporal squama with the inferior border of the parietal bone have been all prematurely fused (arrows) (Figure 4b).

**Figure 5 children-10-01240-f005:**
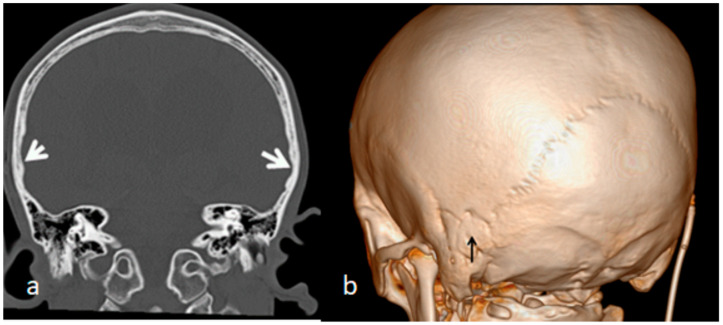
(**a**,**b**). Coronal computed tomography image of the brain demonstrating a local thumb printing/beaten copper appearance (arrows) of the inner cortex in the region signifying bilateral squamosal fusion (white arrows) (Figure 5a). The early sutural fusion extends posteriorly from the pterion and connects the temporal squama with the inferior border of the parietal bone (bulge-like in the lateral view) (arrow heads). Note the early fusion of the mendosal suture (arrow) (known as the accessory occipital suture) and the implanted shunt (Figure 5b).

**Table 1 children-10-01240-t001:** Clinical features of patients in Family I.

Patients	Age	First Presentation	First Diagnosis	History	Intellectual	Skeletal Abnormalities	Other Abnormalities	Diagnosis
Index case	13 years	Hypotonia	Muscular dystrophy	Developmental delay	Poor schooling achievement	Idiopathic kyphoscoliosis	Mitral valve prolapse	SGS
Sibling	7 years	Ocular proptosis	Neonatal thyrotoxicosis	Developmental delay	Poor schooling achievement	Juvenile kyphoscoliosis	---	SGS
Mother	40 years	Adolescent scoliosis	Idiopathic scoliosis at age of 15 years	Multiple spontaneous abortions	Borderline	Idiopathic kyphoscoliosis	Ovarian cysts	SGS
Maternal male sibling	48 years	Seizures at the age of 6 years	Epilepsy	Developmental delay	Borderline	Scoliosis	Aortic root dilation	SGS

## Data Availability

Not applicable.

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
