# Peer review of "Meticulous and Early Understanding of Congenital Cranial Defects Can Save Lives"

_children, 2023, doi:10.3390/children10071240_

Round 1

Reviewer 1 Report

I highly appeciate the difficult topic the authors have selected for their paper. Probably only a few paediatric neurologists, surgeons or neurosurgeons were treated similar cases as presented by the authors. Therefore I highly appreciate the meticulous clinical description of the presented cases and as a general neurosurgeon with some experience in paediatric neurosurgery I learned a lot from the reviewed paper.   

Although the reviewer is not a native speaker of English language, there are some spelling errors that requires correction  - e.g. Miticolous instead of correct Meticulous. The verbal connection Omitting the early closure  sound a little strange. I suggest language editing by a native speaker of English language  also familiar with medical English. 

Author Response

Dear Reviewer,

Thank you so much for your encouraging comments. The process of English editing is ongoing. 

Best regards

Al Kaissi 

Reviewer 2 Report

The Authors in this paper underlying the matter about the identification of Congenital Cranial Defects. They reported the natural history of several unrelated children and adults from two unrelated families.  None of these children underwent cranial sutures assessment and this sort of malpractice was the reason behind a sequence of devastating pathological events that occurred in the lifetime of these children and adults.

After careful review, I would suggest some items to check or improve:

-  I suggest introducing a small overview of the growth of neuro cranium in children (https://doi.org/10.1007/s00381-019-04193-1). In this way, the reader can understand the key concepts of cranial growth.

- in the section “ Conclusion” the authors could propose solutions to avoid misdiagnosing cranial sutures defects. 

- All acronyms need clarification (for example MRI, CT scan, etc..).

- English language needs revision.

Good, minor revision of English. 

Author Response

Dear Reviewer,

Thank you so much for your time. We followed your suggestion and the discussion section has been modified accordingly.

Warmest regards

Al Kaissi

Reviewer 3 Report

The reviewed article presents the results of research on three children and three adults from two unrelated families that exhibited a diverse form of skeletal and visceral abnormalities. A detailed analysis of the history of the examined patients showed, among others, numerous negligence and mistakes made earlier in the process of diagnosis and treatment. According to the authors, 'This sort of malpractice was the reason behind a sequence of devastating pathological events occurred in the life time of these children and adults'. The cognitive value of the article lies in the fact that it emphasizes the importance of a correct early diagnosis for the effective treatment of patients with congenital cranial defects. From this point of view, the article may be of interest to practitioners in the field of neurology and pediatrics.

However, the text of the article contains a number of errors and shortcomings, mainly of a formal and linguistic nature, which should be corrected. Here are some examples:

On the list of institutions to which the authors belong, there is no item marked with the number 5 (next to the name of Franz Grill).

In the 'Summary', change the 'Conclusion' section should be made. In the current version, it basically contains only the objectives of the study, but there are no conclusions resulting from these studies. This way, it does not correspond to the 'Conclusions' section at the end of the article.

The beginning of the section 'Material and methods' should be corrected as it contains sentences that seem unfinished (e.g. 'Clinical documentation and study of several family subjects from two unrelated families.? Multigenerational study of two unrelated families. In total three children (aged 7 -19 year) and three adults from two families (aged 40-52 year)'.?

The text of the article requires a thorough linguistic correction, as it also contains some stylistic and grammatical errors, e.g., p.1, line 36 is 'walker', it should be 'Walker'; p.9, line 321 is 'we confirms', it should be ' we confirm'; p.9, line 339 – remove the number 429.

I support the publication of the article after making the proposed changes and corrections.

The reviewed article presents the results of research on three children and three adults from two unrelated families that exhibited a diverse form of skeletal and visceral abnormalities. A detailed analysis of the history of the examined patients showed, among others, numerous negligence and mistakes made earlier in the process of diagnosis and treatment. According to the authors, 'This sort of malpractice was the reason behind a sequence of devastating pathological events occurred in the life time of these children and adults'. The cognitive value of the article lies in the fact that it emphasizes the importance of a correct early diagnosis for the effective treatment of patients with congenital cranial defects. From this point of view, the article may be of interest to practitioners in the field of neurology and pediatrics.

However, the text of the article contains a number of errors and shortcomings, mainly of a formal and linguistic nature, which should be corrected. Here are some examples:

On the list of institutions to which the authors belong, there is no item marked with the number 5 (next to the name of Franz Grill).

In the 'Summary', change the 'Conclusion' section should be made. In the current version, it basically contains only the objectives of the study, but there are no conclusions resulting from these studies. This way, it does not correspond to the 'Conclusions' section at the end of the article.

The beginning of the section 'Material and methods' should be corrected as it contains sentences that seem unfinished (e.g. 'Clinical documentation and study of several family subjects from two unrelated families.? Multigenerational study of two unrelated families. In total three children (aged 7 -19 year) and three adults from two families (aged 40-52 year)'.?

The text of the article requires a thorough linguistic correction, as it also contains some stylistic and grammatical errors, e.g., p.1, line 36 is 'walker', it should be 'Walker'; p.9, line 321 is 'we confirms', it should be ' we confirm'; p.9, line 339 – remove the number 429.

I support the publication of the article after making the proposed changes and corrections.

Author Response

Dear Reviewer,

Thank you so much for your constructive comments. We almost followed all the points you raised.

Warmest regards

Al Kaissi 

Round 2

Reviewer 2 Report

The authors reported some improvements in the manuscript based on reviews' suggestions.